# Administration of Nicotinamide Mononucleotide (NMN) Reduces Metabolic Impairment in Male Mouse Offspring from Obese Mothers

**DOI:** 10.3390/cells9040791

**Published:** 2020-03-25

**Authors:** Golam M. Uddin, Neil A. Youngson, Sabiha S. Chowdhury, Christopher Hagan, David A. Sinclair, Margaret J. Morris

**Affiliations:** 1Department of Pharmacology, School of Medical Sciences, UNSW Sydney NSW 2052, Australia; golammezbah.uddin@ucalgary.ca (G.M.U.); n.youngson@researchinliver.org.uk (N.A.Y.); s.chowdhury@symbiocelltech.com (S.S.C.); chrisjh95@gmail.com (C.H.); david_sinclair@hms.harvard.edu (D.A.S.); 2Department of Genetics, Paul F. Glenn Laboratories for the Biological Mechanisms of Aging, Harvard Medical School, Boston, MA 02115, USA

**Keywords:** male offspring, maternal obesity, mitochondria, nicotinamide mononucleotide (NMN)

## Abstract

Maternal obesity impacts offspring metabolism. We sought to boost mitochondrial energy metabolism using the nicotinamide adenine dinucleotide (NAD^+^) precursor nicotinamide mononucleotide (NMN) to treat metabolic impairment induced by maternal and long-term post weaning over-nutrition. Male offspring of lean or obese mothers, fed chow or high fat diet (HFD) for 30 weeks post-weaning, were given NMN injection, starting at 31 weeks of age, daily for 3 weeks before sacrifice. Glucose tolerance was tested at 10, 29 and 32 weeks of age to measure short and long term effects of post-weaning HFD, and NMN treatment. Plasma insulin and triglycerides, liver triglycerides and expression of mitochondrial metabolism-related genes were measured at 34 weeks. Impaired glucose tolerance due to maternal and post weaning HFD was significantly improved by only 8 days of NMN treatment. Furthermore, in offspring of obese mothers hepatic lipid accumulation was reduced due to NMN treatment by 50% and 23% in chow and HFD fed offspring respectively. Hepatic genes involved in fat synthesis, transport and uptake were reduced, while those involved in fatty acid oxidation were increased by NMN. Overall this finding suggests short term administration of NMN could be a therapeutic approach for treating metabolic disease due to maternal and post weaning over-nutrition, even in late adulthood.

## 1. Introduction

Prior to the end of the 19th century, health concerns in developed countries were largely focused on poverty, malnutrition, and communicable diseases. However, at the beginning of the 21st century, the World Health Organization (WHO) now considers obesity to be a major health problem [1]. The WHO estimated that overweight and/or obesity results in 2.8 million deaths each year. As with obesity in adults, the prevalence of childhood obesity is rising across the world [2]. Childhood obesity can lead to serious complications and lifelong diseases like cardiac disorders, type 2 diabetes (T2D), psychological complications, pulmonary disorders, and exercise intolerance [3]. Parental overweight (body mass index (BMI) > 25) is considered as a key contributor to the risk of being overweight or obese in early life [4]. Also, it has been reported that early life developmental issues strongly associate with the prevalence of obesity and related metabolic disorders later in life [5,6]. Extensive evidence links increased maternal BMI to adverse offspring metabolic and cardiovascular outcomes, with increased risk for obesity and metabolic disease in offspring [7]. Maternal obesity is associated with non-alcoholic fatty liver disease (NAFLD) in adolescent offspring [8]. Furthermore, mesenchymal stem cells of newborns from obese mothers showed greater lipid accumulation, lower fatty acid oxidation, dysregulation of AMPK (adenosine monophosphate-activated protein kinase) activity and downregulation of genes responsible for fatty acid oxidation [9]. While these long term impacts of maternal obesity for offspring underline the crucial need for intervention, the appropriate intervention for an individual will likely depend on many variables such as gender, age and severity of phenotype.

Mitochondrial dysfunction in muscle and liver is a key mediating mechanism for obesity-related diseases such as insulin resistance and T2D [10,11]. Nutrient oxidation, substrate metabolism, adenosine triphosphate (ATP) production and many other major cellular processes occur in mitochondria. Mitochondrial ATP production occurs through the TCA (tricarboxylic acid) cycle and oxidative phosphorylation (OXPHOS). The reduction of NAD^+^ to NADH, during mitochondrial β-oxidation, glycolysis and the TCA cycle, and the reverse process, oxidation in oxidative phosphorylation, makes NAD^+^ a key component of mitochondrial energy generation [12]. Beyond ATP generation, in recent years’ interest has grown in the role of NAD^+^ as a central signalling molecule and enzyme substrate in many other fundamental biological processes including lifespan regulation, DNA repair, apoptosis, and telomere maintenance [13]. NAD^+^ has been shown to decline with age and under conditions associated with metabolic abnormalities e.g. obesity, diabetes. There has been recent interest in using NAD^+^ boosting agents including nicotinamide mononucleotide (NMN) and nicotinamide riboside (NR) as therapeutic targets for treating metabolic impairments [14,15,16]. It was shown that long term administration of NMN in drinking water successfully reversed age associated body weight gain, improved energy metabolism and insulin sensitivity [17]. Also, we demonstrated that only 18 days of NMN injection improved glucose tolerance, reduced lipid accumulation and upregulated genes responsible for fatty acid metabolism in female offspring of obese mothers [18]. However, no studies to date have examined the effects of NMN in obese male offspring. Investigating both male and female offspring is important due to the differences that have been reported in both human [19] and rodent studies [20,21,22] on the scale of changes in growth and adiposity of offspring. After observing improvements in metabolic performance in female offspring, we were interested to investigate whether short term NMN administration has any effect on male offspring. Further, we were interested in whether similar improvements could be generated even after long-term post-weaning exposure to a HFD that caused extreme obesity. We hypothesised that maternal obesity coupled with longer term post-weaning HFD would worsen the pathological and metabolic conditions in male offspring. Thus the aim of this study was to examine whether under these more chronic conditions, short-term NMN administration could exert beneficial effects in male offspring consuming HFD. 

## 2. Materials and Methods

Animal Experimentation: All animals were housed at the Biological Resources Centre facility, UNSW. Experimental protocols were approved by the Animal Ethics Committee, UNSW, and carried out according to their guidelines and regulations (ethics number 13/25B). The male offspring described here were generated as part of a large maternal obesity cohort that also examined effects of NMN and exercise on their female siblings; Uddin et al. 2017 [18]. To generate offspring, three week old C56BL6/J female mice (*n* = 128) were purchased from Animal Resources Centre, Perth. After a week of acclimatisation mice were assigned to either control chow (*n* = 48; digestible energy 11 kJ/g, 4% of total weight as fat) or HFD (*n* = 80; digestible energy 19 kJ/g, 23.5% of total weight as fat Research Diet #D12451, SF-04-001; Specialty Feeds, Glen Forrest, WA, Australia). At nine weeks of age, females were mated with adult male chow-fed mice from the same source. The pre-pregnancy diet was continued throughout pregnancy and lactation. Dams were singly housed from around 14 days of pregnancy; pups were left undisturbed for the first week to prevent maternal stress. At post-natal day 28 male offspring were weaned and distributed across chow or HFD groups. The experimental timeline is presented in Figure 1A. 

After six weeks of dietary allocation, at 10 weeks of age, a glucose tolerance test (GTT) was conducted to test for changes induced by maternal or post weaning diet induced obesity. Animals from both chow and HFD groups remained on their respective diets for the rest of the experiment. After 25 weeks of HFD feeding, at 29 weeks of age, a GTT was conducted to test for changes induced by maternal or long term post weaning diet induced obesity. Next, pups were further distributed into either vehicle, or NMN groups. Another GTT was performed after 8 days of NMN treatment. NMN was dissolved in PBS and injected i.p. daily for 21 days before sacrifice, 500 mg/kg body weight [14,18,23]. All non-NMN treated mice received a vehicle i.p. injection of PBS daily at the end of the light period. The groups are named by maternal diet, post-weaning diet, NMN/vehicle. Eight groups were generated: Chow-Chow-vehicle: CCV; Chow-Chow-NMN: CCN; Chow-HFD-vehicle: CHV; Chow-HFD-NMN: CHN; HFD-Chow-vehicle: HCV; HFD-Chow-NMN: HCN; HFD-HFD-vehicle: HHV; HFD-HFD-NMN: HHN. Group distributions are shown in Figure 1B. 

Glucose tolerance test: Three GTTs were performed at 10, 29 and 32 week of age. Animals were weighed and fasted for five hours (7am–12pm). Baseline blood glucose was measured by tail nick at time zero (just before glucose bolus). Mice were then challenged with an i.p. glucose bolus (2 g/kg body weight). Blood glucose concentration was measured using an Accu-ChekH glucose meter (Roche Diagnostics, Nutley, USA) at 15, 30, 60, 90, 120 and 180 min after glucose administration.

Sample collection, processing and plasma insulin concentration: At 34 weeks NMN or vehicle was injected four hours before anaesthetic. Animals were weighed and fasted for five hours then anesthetized (ketamine/xylazine 200/20 mg/kg, i.p.). A blood sample was collected by cardiac puncture and plasma collected after centrifugation (2000 *g*-10 min). Plasma insulin concentrations were measured by Ultra-Sensitive Mouse Insulin ELISA kit (Crystal Chem Inc., Elk Grove Village, IL, USA). After decapitation, retroperitoneal fat, quadriceps muscle and liver were weighed. Liver was ground using a tissue pulverizer on dry ice and liquid N2.

Hepatic and plasma triglyceride content: Triglyceride measurement was conducted according to [23] in plasma (collected just before cull) or liver after tissue homogenisation, using a commercially available colorimetric assay kit, TG reagent Triglyceride GPO-PAP (cat# 11730711 216 Roche/Hitachi). After incubation samples were read on a Bio-Rad iMark plate reader.

Quantitative RT-PCR: RNA was extracted using Tri-reagent (Sigma, St. Louis, MO, USA), DNA that may have been co-extracted was removed by the addition of a DNase treatment (Qiagen). Final RNA quality and concentrations were determined using a Biospec-nano spectrophotometer (Shimadzu Biotech, Nakagyo-ku, Kyoto, Japan). Candidate genes were tested to investigate which molecular pathways may be involved in generating the observed effects in fat and glucose metabolism (primer sequences are listed in Appendix A). These genes were chosen based on our previous work examining effects of NMN and exercise in female offspring [18]. Primer-coated custom designed rtPCR plates were purchased from Bio-Rad, Australia. All sample PCRs were performed in duplicate, and all genes of interest were normalized by dividing by the geometric mean of two control genes Gapdh and Ywhaz. No difference in expression of housekeeper genes was observed across treatment groups. Relative gene expression methods were calculated using the 2-(∆∆CT) method.

Statistical analysis: Data are expressed as mean ± SEM. Body weight and GTT were analysed by three-way mixed ANOVA. Three way ANOVAs were used to compare all 8 groups with maternal diet, post-weaning diet and NMN treatment as factors, to examine main effects. Two-way ANOVAs were used to analyse the effects of the offspring from obese or lean mothers. If data were not normally distributed they were log transformed to achieve normality before they were analysed. Different superscripts represent significant differences between the designated groups (^†^ maternal diet effect; * Post-weaning diet Effect; ^ NMN effect). Differences were considered significant at *p* < 0.05.

## 3. Results

### 3.1. Maternal Diet had a Significant Impact on Pup Body Weight 

Pups from mothers consuming HFD were heavier than those from lean mothers (Figure 2A). The impact of maternal obesity was exacerbated in those pups eating HFD; a significant interaction between maternal and post-weaning diet was observed. As expected, post-weaning HFD increased body weight regardless of maternal diet. The pattern over time is similar to what we observed in female offspring [18]. The male offspring of obese mothers were significantly heavier than those from lean mothers regardless of their post weaning diet. Liver and retroperitoneal fat mass but not muscle mass were also significantly increased due to maternal HFD. When tissue weight was standardised as % body weight, significant maternal diet effects were observed in liver, quadriceps muscle and retroperitoneal fat. 

### 3.2. Consumption of HFD Increased Final Body Weight and Tissue Mass with a Recovery Phenotype by NMN

As expected, consumption of HFD post-weaning increased final body weight as well as liver, muscle and fat mass (Table 1); when tissue weight was standardised as percent body weight, these significant differences remained. After 30 weeks of post-weaning HFD, three weeks of NMN treatment significantly reduced body weight but this was only observed in male offspring of lean mothers who consumed a HFD (47.2 ± 1.0 g vs. 41.8 ± 0.9 g; *p* < 0.001 Table 1). The short term NMN treatment had no significant effect on body weight on offspring from obese mothers. Interestingly retroperitoneal fat mass was also reduced by 3 weeks of NMN treatment in offspring consuming HFD regardless of maternal diet. When quadriceps muscle weight was standardised as percent body weight, it was significantly increased by NMN in the offspring from obese mothers consuming HFD, and a similar trend was observed in the offspring from lean mothers (*p* < 0.058, Table 1), probably reflecting a lower overall fat mass in these animals.

### 3.3. The Effects of Maternal and Post-weaning Diet on Glucose Tolerance at 10 Weeks of Age

The effects of maternal and post-weaning diet on glucose tolerance over time are shown in Figure 3A. A significant effect of post-weaning HFD was observed in glucose clearance (Figure 3A). At this early time point (10 weeks of age), no effect of maternal diet on GTT was evident in offspring consuming chow (Figure 3A,B), however a reduction in glucose tolerance was seen in those consuming HFD. Specifically, blood glucose remained higher at 60, 90, 120 and 180 min compared to offspring from lean mothers. This suggests male offspring from obese mothers were less glucose tolerant. The area under the GTT curve (AUC) at 180 minutes is presented in Figure 3B; both maternal diet and post-weaning diet effects were observed, in line with the significantly greater glucose response induced by HFD.

### 3.4. The Effects of Maternal and Post-weaning Diet on Glucose Tolerance at 29 Weeks of Age

At 29 weeks of age, both maternal and long term post-weaning HFD had significant negative impacts on plasma glucose clearance (Figure 3C), with a worsening of glucose handling compared to the earlier time point. Analysis revealed animals consuming HFD post-weaning over this longer term were markedly more glucose intolerant relative to those consuming chow, regardless of their maternal status. Also, in contrast to the 10 week GTT, here at 29 weeks there was a clear separation of glucose curves in chow-fed offspring from obese mothers. The area under the GTT curve at 180 minutes (Figure 3D) support these findings.

### 3.5. NMN Treatment Was Beneficial Only for the Most Metabolically Compromised Mice Group

Finally, to examine if the impaired glucose tolerance due to long-term HFD consumption can be improved in the short term by NMN treatment (eight days), a GTT was conducted in all eight groups at 32 weeks of age. As shown in Figure 4A–D the effect of NMN on glucose tolerance appeared to vary with the maternal dietary status. Surprisingly, NMN impaired glucose tolerance in mice consuming chow. This was reflected in the AUC data (Figure 4B) with a significant 40% increase in CCN versus CCV mice. In offspring of lean mothers consuming HFD, NMN had no additional impact. 

Offspring of obese mothers are depicted in Figure 4C,D. In this case the reduction in glucose tolerance in mice consuming HFD post-weaning was significantly improved in the group receiving NMN injection. On the other hand, in offspring of obese mothers who consumed chow post-weaning, there was a significant increase in blood glucose concentrations, albeit briefly, in the mice receiving NMN (Figure 4C) as reflected in a greater AUC (Figure 4D). Overall, it appears that NMN was beneficial in those that were most metabolically compromised.

To explore the effects on maternal as well as post-weaning diet and NMN treatment on fasting blood glucose concentrations, data from the zero-time point prior to the glucose bolus are shown in Table 1. At 10 weeks of age, there were significant effects of post-weaning diet as well as maternal diet. At 29 weeks, there was no significant impact of maternal diet (although glucose tended to be increased in those from obese mothers), and post-weaning HFD diet significantly increased basal glucose levels (Table 1). In addition, NMN treatment was associated with greater resting blood glucose in those offspring consuming chow. No differences were observed in those consuming HFD (Table 1). 

### 3.6. Impacts of HFD and NMN Treatment on Insulin

Plasma insulin concentrations at sacrifice are presented in Table 1. Post-weaning HFD consumption was associated with significantly higher plasma insulin in offspring compared to those consuming chow diet, regardless of their maternal diet. As expected, maternal HFD also had a significant effect on insulin levels, but only in offspring consuming chow. However, a significant increase in plasma insulin concentration was observed in obese offspring from HFD consuming mothers. NMN treatment was associated with increased plasma insulin concentrations at the end of the study, particularly in the offspring consuming HFD from obese mothers (Table 1). 

### 3.7. Hepatic Triglyceride Was Increased Due to HFD Consumption and Recovered by NMN Treatment

Quantitation of both liver triglyceride content and plasma triglyceride concentration shows an interesting graded response whereby both maternal and post-weaning HFD led to increased fat accumulation in male offspring, with greater impacts of the combination (Figure 2B,C). A strong main effect of intervention was observed, suggesting that three weeks of NMN treatment significantly reduced triglyceride accumulation in the liver and plasma of male offspring from obese mothers regardless of their post-weaning diet. 

### 3.8. Changes in Gene Expression Responsible for Glucose and Fatty Acid Metabolism

Next the hepatic expression of a range of genes that may be implicated in the impact of diet and NMN on glucose tolerance, adiposity and triglyceride accumulation were examined (Figure 5 and Table 1). Among those, Acc1 (acetyl-CoA carboxylase 1), Cd36, a fatty acid transporter, Fabp (Fatty acid binding protein) and Fasn (fatty acid synthase) mRNA expression was significantly increased by post-weaning HFD in offspring regardless the maternal diet. This suggests HFD consumption increases fatty acid synthesis and fat transport into the liver. Mitochondrial pyruvate carrier 1 (Mpc1) and Pparγ were increased due to HFD consumption in the offspring from lean mothers. Cpt1, responsible for transporting fatty acyl CoA, Hadh a marker for mitochondrial β oxidation and Pgc1α, a marker for mitochondrial biogenesis mRNA expression levels were significantly increased by HFD consumption in the offspring of obese mothers only (Table 1). There was also an overall maternal diet effect on the mRNA expression of Cd36, Fabp and Pparγ. NMN treatment did not affect hepatic mRNA expression significantly in any offspring consuming chow regardless of maternal diet. However, in offspring consuming HFD from obese and lean mothers, NMN significantly reduced the mRNA expression of Acc1, Mpc1 and Cd36. Furthermore, Cpt1, Fabp and Fasn mRNA expression was reduced significantly by NMN treatment in mice consuming HFD from obese mothers. Overall, these results indicate that even short term NMN treatment has potential reduce gluconeogenesis in the most obese mice i.e. those exposed to maternal obesity and 30 weeks of post-weaning HFD consumption. Future detailed studies to investigate gluconeogenesis are warranted to confirm this. Interestingly, Pparg and Pgc1a expression was increased by NMN treatment only in mice consuming HFD from lean mothers. Future work is needed to determine whether these liver gene expression changes explain the reduction of adiposity and improved glucose intolerance in the obese mice. 

## 4. Discussion

Maternal obesity has been shown to impair metabolic profile in late adulthood [24,25,26]. Furthermore, these impacts are known to be exacerbated by post-weaning HFD consumption [18,27,28], and in this study mice exposed to both insults, maternal obesity and chronic post-weaning HFD, were twice as heavy as CCV controls. In this study, we wanted to expose male offspring to longer term post-weaning HFD, which as expected, after 30 weeks led to dramatic increases in body weight compared to offspring consuming chow from both lean (CHV 84% heavier than CCV) and obese (HHV 88% heavier than HCV) mothers (Table 1). We found that only three weeks of NMN administration was able to reduce fat mass, liver and plasma triglyceride levels, and to improve glucose tolerance in mice with severe metabolic impacts due to the combination of maternal obesity and chronic high fat diet consumption. The benefits of NMN appear to have been partly mediated by the modulation of genes with roles in fat transport and metabolism in the liver.

An increasing body of evidence suggests that two major NAD^+^ precursors, NR and NMN could offer therapeutic potential for metabolic disease and ageing [29]. NR supplementation in food (200–400 mg/kg, ranging from 8–18 weeks) has been shown in multiple studies with HFD fed male mice to protect against body weight gain, to improve glucose tolerance, and to reduce hepatic steatosis, [16,30], lipid accumulation, liver fibrosis [31], and hepatic ER stress [32]. Various rodent studies that used single or multiple NMN doses (range 300–500 mg/kg) have shown beneficial effects in cardiac injury [33], heart failure [34], improved brain function and memory [35], enhanced mitochondrial oxidative metabolism [36] and hepatic mitochondrial respiration [15]. Furthermore, fructose-rich-diet fed male mice showed improved insulin secretion with a single NMN dose [37]. Additionally, improved glucose tolerance and insulin sensitivity were shown with 7 and 11 days of NMN administration in HFD-fed and age-induced diabetic male mice respectively [14]. Previously we have shown that short term administration of NMN through i.p. injection (500 mg/day/kg for 17 day) was as beneficial as six weeks of exercise in obese mice [23], and 18 days of NMN injection could improve metabolic impairments in female offspring of obese mothers [18]. In comparison to this, other pharmacological approaches i.e. leucine, metformin, and exendin-4, superoxide dismutase mimetic and peroxynitrite supplementation for 5–8 weeks have been shown to reduce circulating and hepatic triglyceride accumulation in HFD fed obese rats and mice [38,39]. Head to head comparisons between NAD^+^ intermediates and these other drugs is an important future area of research, as is determining how long the benefits of NAD^+^ intermediates persist after termination of administration. 

The results of the current study extend previous work highlighting potential beneficial metabolic effects of NMN administration in obesity, induced by maternal and/or post-weaning overfeeding [18,23]. These benefits include improved glucose tolerance, reduced adiposity and lower hepatic triglyceride accumulation. In our previous NMN studies adult female mice received HFD for 12 weeks [23] or 18 weeks in combination maternal HFD [18]. It is interesting to note that in the current study only 3 weeks of NMN treatment was able to reduce triglyceride concentrations significantly even with the longer period of post-weaning HFD exposure (30 weeks). The reduction in hepatic triglyceride concentration of offspring consuming HFD was only present in those from obese mothers, possibly indicating that NMN is more effective in ameliorating programmed liver triglyceride accumulation than solely diet-induced accumulation. Recently Dall and colleagues reported that the NAD^+^ salvage pathway is unaffected by prolonged HFD consumption [40]. Another study with skeletal muscle overexpression of a rate limiting enzyme of the NAD^+^ salvage pathway resulted in increased muscle NAD^+^ concentration and exercise endurance [41]. Therefore, as we expect that the changes in liver triglycerides are driven by increased NAD^+^ in that organ [18] it is attractive to hypothesise that the NAD^+^ salvage pathway is reduced by maternal obesity and can be ameliorated by the administration of NMN. 

Here, we investigated three possible reasons for reduced liver triglyceride stores; increased fat catabolism, fat synthesis or reduced fat uptake. Similar to our findings in their female siblings [18] NMN induced gene expression changes consistent with a reduction in fat import into the liver (Cd36) and less fatty acid synthesis (Fasn, Acc1). Again, as was observed in females, reductions in both Acc1 and Fasn by NMN in the HFD-fed groups suggest that altered fatty acid synthesis may also contribute to the reduced liver triglycerides. It is unclear whether liver fatty acid catabolism is affected as NMN significantly reduced the mRNA expression of Cpt1 suggesting less fatty acyl CoA transportation into the mitochondria, but no significant change in the mitochondrial β-oxidation gene Hadh was seen. However, to understand the exact mechanism, if obesity increases TG accumulation by downregulation of fatty acid oxidation, and providing NMN can rescue that phenotype, it will be important to measure directly fatty acid oxidation rates. This current study was not designed to measure the oxidation rates using fresh tissue, however, future studies are warranted to find out the exact effect.

These data provide additional evidence that short term administration of NMN improves glucose tolerance in obese mice [14,18,23], as observed in the heaviest HHN group at 32 weeks of age. However, at this time NMN appeared to increase blood glucose during GTT in chow fed mice, regardless of maternal diet; CCN and HCN relative to CCV and HCV, respectively. To our knowledge this has not been previously reported. This may suggest that NMN is of greater benefit under conditions of extreme obesity rather than in individuals undergoing healthy aging. It is not clear whether these findings relate to the bolus administration of NMN, as long term administration of NMN (12 months) in drinking water was shown to improve glucose tolerance in chow-fed older mice [17]. The differences in glucose tolerance may be explained by an improvement in the sensitivity of pancreatic β-cells to glucose and therefore increased insulin secretion in response to a glucose bolus. This explanation is supported by another study that showed that NMN supplementation improved glucose stimulated insulin secretion in isolated β-cells from fructose-fed mice that had pancreatic islet dysfunction [37]. Alternatively, a recent study showed that insulin resistance in adipocyte specific Nampt knockout mice was improved by NMN treatment [42]. Therefore, NMN may alter glucose tolerance through changing tissue sensitivity to insulin and/or beta-cell response to glucose, and further work should test these possibilities. 

Conducting GTT at two time points, 10 and 29 weeks of age, enabled us to explore the impact of maternal obesity on glucose tolerance over time. Others reported that there were no maternal HFD impacts on glucose levels when male mice offspring consumed chow for 23 weeks [43]. In our cohort, male offspring at 10 weeks of age consuming chow did not manifest any impact of maternal HFD on the GTT, and neither did their female siblings at 18 weeks of age [18]. However, at 29 weeks of age, the chow consuming male offspring from obese mothers had clearly reduced glucose tolerance. These observations point to latent effects of maternal obesity that are revealed with aging. This interaction between maternal diet and offspring age could be of clinical significance in human populations due to the recent growth in maternal obesity and the concurrent lengthening of average lifespan. Even children of obese mothers that eat a healthy diet may have increased disease risk in later life [44].

Finally, comparison of these data from males with their female siblings from our previous study [18] allowed us to investigate sex-specific responses to maternal obesity and post weaning HFD. The female siblings underwent 15 weeks of post weaning HFD [18]. After 15 weeks of post weaning HFD males were 33% (CHS vs. CCS) and 55% (HHV vs. HCV) heavier (Figure 2), whereas at that time the equivalent female offspring group weights were 15% (CHS vs. CCS) and 44% (HHS vs. HCS) different [18]. Therefore, our studies add to others which suggest that male offspring of obese mothers may be more prone to obesity than female offspring [45,46]. However, more work is needed to evaluate the relative sensitivities of male and female offspring to maternal obesity as other studies have reported greater effects in females [19]. 

## 5. Conclusions

Overall, our data clearly highlight the negative effects of maternal and long term post-weaning HFD on male offspring, and point to beneficial metabolic outcomes induced by a short period of NMN administration. These data support the use of NMN as a potential method to combat the intergenerational consequences of the obesity epidemic. 

## Figures and Tables

**Figure 1 cells-09-00791-f001:**
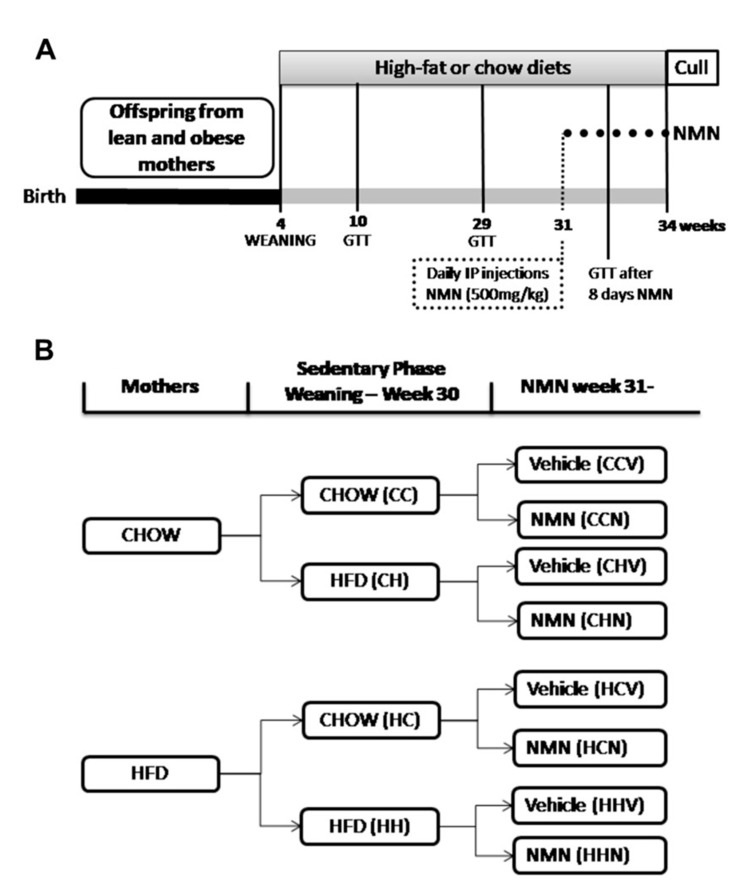
Panel (**A**) shows the timeline of the experiment: Male offspring of lean and obese mothers were assigned to chow or HFD at 4 weeks of age. GTTs were performed after 6 and 25 weeks of dietary intervention (10 and 29 weeks of age). At 31 weeks of age, daily i.p. injection of NMN (500 mg/kg body weight) or vehicle commenced. GTT was carried out after 8 days. All non-treated animals were exposed to daily vehicle (PBS) injection. Injections occurred daily for 21 days preceding cull. Panel (**B**) describes the experimental groups: Male offspring were assigned post-weaning diets of chow or HFD, generating 4 groups. At 31 weeks of age offspring received either vehicle or NMN across 8 different groups (all *n* = 12) Chow-Chow-vehicle: CCV; Chow-Chow-NMN: CCN; Chow-HFD-vehicle: FCHV; Chow-HFD-NMN: CHN; HFD-Chow-vehicle: HCV; HFD-Chow-NMN: HCN; HFD-HFD-vehicle: HHV; HFD-HFD-NMN: HHN CCV. The letters refer to maternal diet, offspring diet and treatment respectively.

**Figure 2 cells-09-00791-f002:**
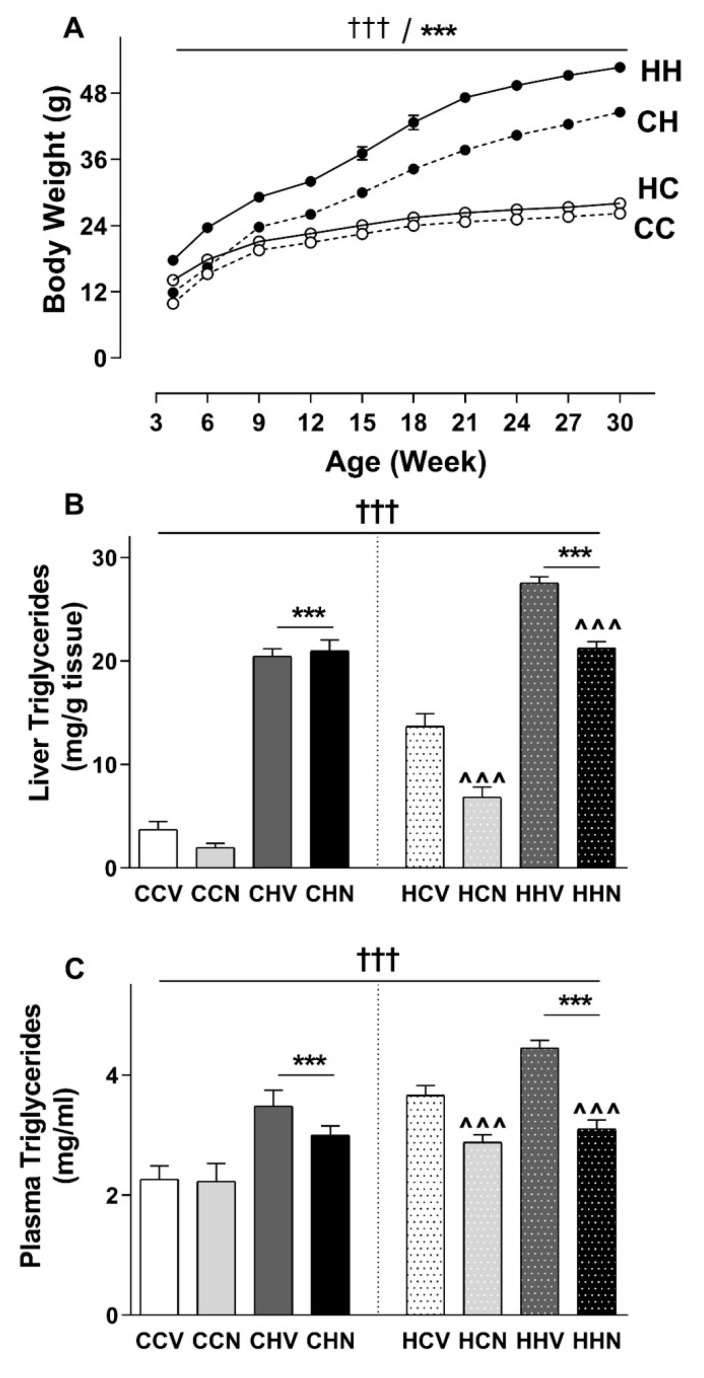
Panel (**A**) shows maternal diet and post-weaning diet effects on body weight of male offspring from 4 to 30 weeks of age: Data are shown as mean ± SEM, *n* = 9–24 per group. The first letter represents the maternal and the second letter represents post-weaning diet; chow (C) or HFD (H) of CC (open circle dotted line), CH (closed circle dotted line), HC (open circle solid line) and HH (closed circle solid line) groups. Data were analysed by repeated two-way ANOVA. The significant effects (simple main effects) are presented as: *** *p* < 0.001 post-weaning diet effect; ^†††^
*p* < 0.001 maternal diet effect. Panel (**B**) and (**C**) shows maternal and post-weaning diet, and NMN treatment effects on triglyceride levels of male offspring. Data are shown as mean ± SEM, *n* = 9–12 per group. The first letter represents the maternal diet and the second letter post-weaning diet; chow (C) or HFD (H); the third letter represents treatment; vehicle (V) or NMN (N). To explore maternal and post-weaning diet effects, data were analysed by three-way ANOVA with maternal diet, post-weaning diet, and NMN as factors. The significant main effects of maternal diet are presented as: ^†††^
*p* < 0.001 maternal diet effect; *** *p* < 0.001 post-weaning diet effect and ^^^ *p* < 0.001 NMN effect.

**Figure 3 cells-09-00791-f003:**
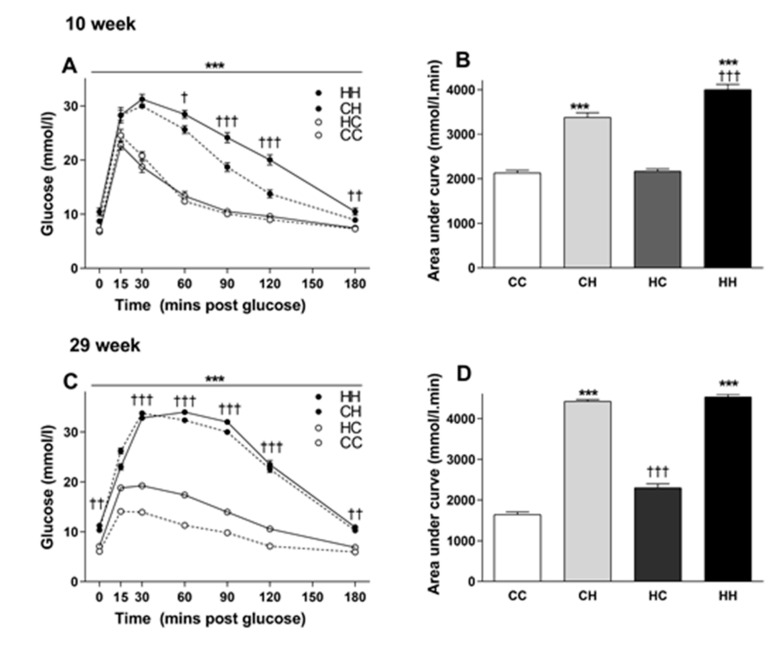
GTT in male offspring of lean and obese mothers at 10 and 29 weeks of age (**A**–**D**): Data are shown as mean ± SEM, *n* = 11–12 per group. The first letter represents the maternal diet; chow (C, dotted lines) or HFD (H, solid lines). The second letter represents post-weaning diet; chow (**C**, open circle) or HFD (H, closed circle). (**A**) Blood glucose concentration over time and (**B**) AUC at 10 weeks of age. Long term maternal and post-weaning diet effects on GTT were examined in offspring at 29 weeks of age (**C**,**D**). Data were analysed by repeated two-way ANOVA. The significant effects (simple main effects) are presented as: ^†^
*p* < 0.05; ^††^
*p* < 0.01; ^†††^
*p* < 0.001 maternal diet effect; *** *p* < 0.01 post-weaning diet effect.

**Figure 4 cells-09-00791-f004:**
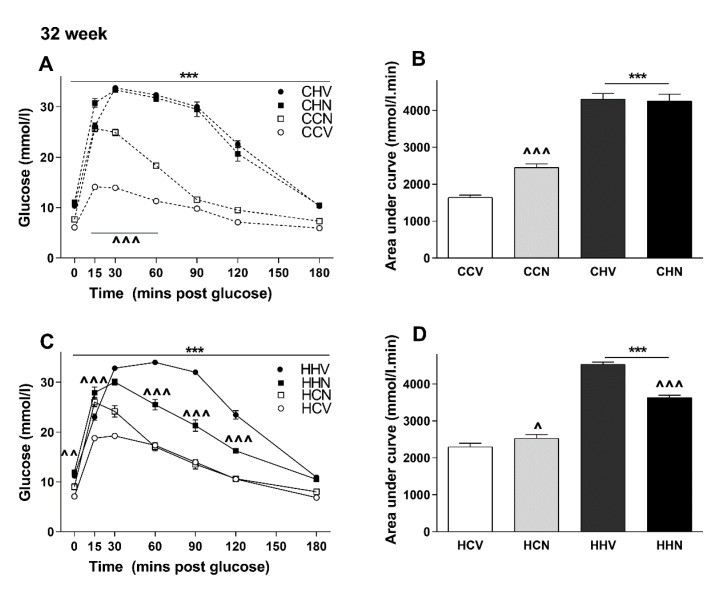
GTT in male offspring of lean and obese mothers following 8 days of NMN treatment at 32 weeks of age (**A**–**D**): Data are shown as mean ± SEM, *n* = 11–12 per group. The first letter represents the maternal diet; chow (C, dotted lines) or HFD (H, solid lines). The second letter represents post-weaning diet; chow (C, open circle) or HFD (H, closed circle). The third letter represents vehicle (V) or NMN (N, squares). Effects of NMN treatment on GTT in offspring from lean (**A**,**B**) and obese (**C**,**D**) mothers were analysed at 32 weeks of age by separate ANOVAs. The significant effects (simple main effects) are presented as: ^†^*p* < 0.05; ^††^*p* < 0.01; ^†††^*p* < 0.001 maternal diet effect; *** *p* < 0.01 post-weaning diet effect; ^ *p* < 0.05, ^^ *p* < 0.01, ^^^ *p* < 0.001 NMN effect.

**Figure 5 cells-09-00791-f005:**
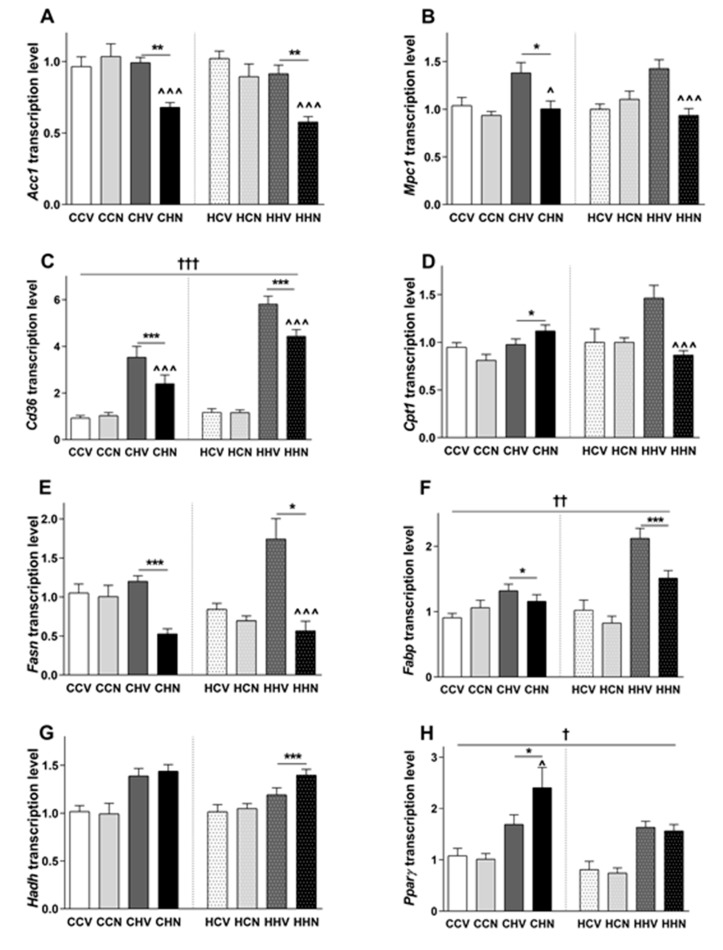
Effect of NMN on expression of genes involved in mitochondrial biogenesis, fat transport and gluconeogenesis in liver. Data are shown as mean ± SEM, *n* = 9–10 per group. The first letter represents the maternal diet and the second letter post-weaning diet; chow (C) or HFD (H); the third letter represents treatment; vehicle (V) or NMN (N). To explore the maternal and post-weaning diet effect on mRNA expression of Acc1 (**A**), Mpc1 (**B**), Cd36 (**C**), Cpt1 (**D**), Fasn (**E**), Fabp (**F**), Hadh (**G**) and Pparγ (**H**) were analysed by three-way ANOVA with maternal diet, post-weaning diet, and NMN as factors. The significant main effects of maternal diet are presented as: ^†^
*p* < 0.05, ^††^
*p* < 0.01, ^†††^
*p* < 0.001 maternal diet effect. To explore the impact of NMN on these parameters, separate two-way ANOVAs were performed in the offspring of lean (left) or obese (right) mothers that consumed chow or HFD. The significant effects (simple main effects) are presented as: * *p* < 0.05, ** *p* < 0.01, *** *p* < 0.001 post-weaning diet effect; ^ *p* < 0.05, ^^^ *p* < 0.001 NMN effect.

**Table 1 cells-09-00791-t001:** Body weight, organ mass, fasting insulin and blood glucose concentrations, and Pgc1 expression of 34-week old male offspring of obese mothers that consumed chow or HFD, under control or NMN treatment. Fasting glucose taken at 10 and 32 weeks of age. At the 10-week time point mice had not been separated into treatment groups. Data are shown as mean ± SEM, n = 9–12 per group. The first letter represents the maternal diet and the second letter post-weaning diet; chow (C) or HFD (H); the third letter represents treatment; vehicle (V) or NMN (N). To explore maternal and post-weaning diet effects, data were analysed by three-way ANOVA with maternal diet, post-weaning diet, and NMN as factors. The significant main effects of maternal diet are presented as: † *p* < 0.05, †† *p* < 0.01, ††† *p* < 0.001 maternal diet effect. To explore the impact of NMN on these parameters, separate two-way ANOVAs were performed in the offspring of lean (left) or obese (right) mothers that consumed chow or HFD. The significant effects (simple main effects) are presented as: * *p* < 0.05, ** *p* < 0.01, *** *p* < 0.001 post-weaning diet effect; ^ *p* < 0.05, ^^ *p* < 0.01, ^^^ *p* < 0.001 NMN effect, NA—Not applicable.

	Offspring from Lean Mothers	Offspring from Obese Mothers	Maternal Diet
CCV	CCN	CHV	CHN	HCV	HCN	HHV	HHN
Final Body Weight (g)	25.6 ± 0.4	25.2 ± 0.3	47.2 ± 1.0 ***	41.8 ± 0.9 ^^^	27.8 ± 0.5	26.8 ± 0.6	52.4 ± 1.1 ***	51.0 ± 1.4	†††
Liver mass (mg)	945 ± 55	952 ± 35	1668 ± 136 ***	1425 ± 89	1059 ± 75	935 ± 36	2603 ± 170 ***	2456 ± 198	†††
Quadriceps mass (mg)	287 ± 13	311 ± 13	329 ± 11 *	335 ± 12	322 ± 3	302 ± 9	333 ± 15 ***	361 ± 14	
RP mass (mg)	84 ± 9	108 ± 11	639 ± 19 ***	520 ± 15 ^^^	201 ± 19	195 ± 22	729 ± 44 ***	633 ± 29 ^	†††
Liver (% BW)	3.68 ± 0.20	3.79 ± 0.12	3.68 ± 0.22	3.38 ± 0.15	3.78 ± 0.22	3.49 ± 0.10	4.96 ± 0.28 ***	4.52 ± 0.24	†††
Quadriceps (% BW)	1.12 ± 0.05	1.22 ± 0.06	0.69 ± 0.02 ***	0.81 ± 0.03	1.15 ± 0.03	1.11 ± 0.02	0.57 ± 0.07 ***	0.71 ± 0.04 ^	†
RP (% BW)	0.33 ± 0.04	0.39 ± 0.03	1.34 ± 0.03 ***	1.25 ± 0.06	0.71 ± 0.06	0.71 ± 0.07	1.41 ± 0.08 ***	1.24 ± 0.05	†††
Basal glucose (mmol/L)Before NMN at 10^th^ week	7.05 ± 0.23	NA	8.72 ± 0.31 **	NA	6.75 ± 0.27	NA	10.42 ± 0.70 ***	NA	††
Basal glucose (mmol/L)After NMN at 32^nd^ week	6.08 ± 0.17	7.68 ± 0.31 ^^	10.34 ± 0.22 ***	11.05 ± 0.45	7.08 ± 0.12	9.01 ± 0.60 ^^	11.22 ± 0.28 ***	11.94 ± 0.30	
Plasma Insulin (ng/mL)	0.15 ± 0.02	0.23 ± 0.05	2.10 ± 0.63 ***	3.02 ± 0.48	0.45 ± 0.09	0.64 ± 0.08	1.57 ± 0.25 ***	8.23 ± 1.30 ^^^	†††
*Pgc1* gene expression	0.94 ± 0.06	0.95 ± 0.13	0.72 ± 0.07	1.18 ± 0.23 ^	1.03 ± 0.14	1.01 ± 0.09	0.58 ± 0.04 ***	0.63 ± 0.04

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
