# Peer review of "Administration of Nicotinamide Mononucleotide (NMN) Reduces Metabolic Impairment in Male Mouse Offspring from Obese Mothers"

_cells, 2020, doi:10.3390/cells9040791_

Round 1
Reviewer 1 Report
In this manuscript, the authors reported the metabolically beneficial
effects of NMN in the obese mouse models induced by maternal obesity, HFD or both. The authors first seemed to be interested in phenotypes derived from mother obesity, but NMN generally exerted similar effects on mice with obesity. This is an interesting paper, but several points are raised in order to gain some insights into the underlying mechanisms.
- It seems critical whether NMN improved insulin sensitivity, insulin secretion, or both. Since liver steatosis was ameliorated, one might expect insulin sensitivity was improved, but serum insulin levels were significantly higher in HHN groups, which may suggest NMN directly improved pancreatic beta cell function. Therefore, the authors are advised to assess insulin secretion and sensitivity separately. How about the glucose-responsive insulin secretion during OGTT? How about insulin sensitivity by insulin tolerance test?
- Related to the above point, the reason why the NMN worsened glucose tolerance in chow groups (lean subjects) should be discussed, since this would be important from the practical point of view.
- Hepatic TG content and serum TG levels were reduced in obese group,and the authors examined genes about fat synthesis and transport. Sinec NMN can activate mitochondrial function, fat oxidation should be more directly assessed.
- Results 3.6: Gluconeogenesis is one of the key issues related to insulin sensitivity, but it is difficult to conclude from Figure 5 that NMN reduced gluconeogenesis in obese mice, since genes of gluconeogenesis pathway (such as Pepck and G6Pc) were not assessd.
【Minor points】
- Abstract: It is difficult to understand from this version of Abstract that NMN was started at 31-week old.
- Results 3.6: There are several letters missing (such as Ppar and Pge1).
- Figure 4: Figure legend said NMN was administered for 3 weeks, but actually OGTT here was performed only eight days of NMN.
- In the text, the authors stated NMN reduced Pparg and Pgc1a mRNAsin CHN group, but they seemed increased in Fig 5H and Table1.
Author Response
We thank the reviewers for their constructive criticisms related to our paper. We have now addressed all of the issues raised by the reviewers in the revised the manuscript. The modifications are highlighted in the revised version (yellow highlight). Related references for the responses to the reviewers are listed at the end of this document following the MDPI-Cells citation style.

Reviewer 2 Report
In the present manuscript, Mezbah Uddin et al investigate the metabolic effects of nicotinamide mononucleotide (NMN), a precursor of nicotine adenine dinucleotide (NAD+) in transgenerational obesity. The authors use offspring (F1) of mice derived from diet-induced obese mothers (P) or regular chow diet-fed mice. They also feed a regular or chow diet F1 from week 4 followed by NMN treatment at 31 weeks and for 4 weeks to then assess whether NMN improves different metabolic parameters. Mezbah et al find that NMN reduces liver and plasmatic triglycerides, improves glucose tolerance and increases fatty acid oxidation while reduces fatty acid synthesis in the liver.
Overall, the study is very well designed, conducted and controlled. This leads to clear results and interpretations. The effects of NMN on improved metabolism of the experimental mice are solid. Even though there is a lack of a mechanism to explain this phenotype, the results are very convincing. It opens perspectives for future research to understand the effect of NMN on the transgenerational inheritance of metabolic malapdations.
There is however a minor concern. The authors claim in figure 5 a reduction in gluconeogenesis, they show some genes who claim to be important in gluconeogenesis. The canonical gluconeogenic genes Pepck and Glucose-6-phosphatase should be measured. To claim there is a decreased gluconeogenesis, a pyruvate tolerance test should also be performed. I suggest to either include these experiments or in alternative reduce the power on the statement that the gluconeogenesis is reduced.
Author Response

(The authors gave the same response as above.)

Round 2
Reviewer 1 Report
The authors responded to each of my previous comments sincerely. Although they did not perform additional experiments, the problems I raised are now well taken in Discussion. Only two minor points are suggested.
- According to my previous point 2 (NMN worsened glucose tolerance in lean groups), I still think the assessment of insulin secretion after NMN treatment are most critical in these groups, and this point should be added as one of the future study targets in the corresponding paragraph in Discussion.
- The newly added sentences (page 15, paragraph 03, line 450-457): “insulin sensitivity and/or response to insulin” may be read as “insulin sensitivity and/or insulin response to glucose”, I am afraid.
Author Response
Response to Reviewer comments:
We thank the reviewer again for the constructive criticism related to our paper. Along with our previous responses, we have now addressed the two minor issues raised by the reviewer in the revised the manuscript. The modifications are highlighted in the revised version (yellow highlight).
Reviewer #1
The authors responded to each of my previous comments sincerely. Although they did not perform additional experiments, the problems I raised are now well taken in discussion. Only two minor points are suggested.
According to my previous point 2 (NMN worsened glucose tolerance in lean groups), I still think the assessment of insulin secretion after NMN treatment are most critical in these groups, and this point should be added as one of the future study targets in the corresponding paragraph in discussion.
The newly added sentences (page 15, paragraph 03, line 450-457): “insulin sensitivity and/or response to insulin” may be read as “insulin sensitivity and/or insulin response to glucose”, I am afraid.
Response: We agree with the reviewer that this is a crucial experiment. We have acknowledged it in our previous response as well and added additional explanation. Regarding the added sentence we have now modified the sentence as per the reviewer’s suggestion.
Insert, page 15, paragraph 03, line 455-457
“Therefore, NMN may alter glucose tolerance through changing tissue sensitivity to insulin and/or beta-cell response to glucose, and further work should test these possibilities.”